# Molecular Characterization of White Wines Antioxidant Metabolome by Ultra High Performance Liquid Chromatography High-Resolution Mass Spectrometry

**DOI:** 10.3390/antiox9020115

**Published:** 2020-01-28

**Authors:** Remy Romanet, Florian Bahut, Maria Nikolantonaki, Régis D. Gougeon

**Affiliations:** Université de Bourgogne Franche-Comté, AgroSup Dijon, PAM UMR A 02.102, Institut Universitaire de la Vigne et du Vin, Jules Guyot, Rue Claude Ladrey, BP 27877, 21078 Dijon Cedex, France; remy.romanet@gmail.com (R.R.); bahut.florian@gmail.com (F.B.); regis.gougeon@u-bourgogne.fr (R.D.G.)

**Keywords:** UHPLC-QqTOF-MS, untargeted analysis, thiols, Chardonnay wine oxidation, nucleophiles, peptides

## Abstract

The knowledge about the molecular fraction contributing to white wines oxidative stability is still poorly understood. However, the role of S- and N-containing compounds, like glutathione and other peptides, as a source of reductant in many oxidation reactions, and acting against heavy metals toxicity, or lipid and polyphenol oxidation as ROS-scavenger is today very well established. In that respect, the aim of the present study is to introduce an original analytical tool for the direct determination of the available nucleophilic compounds in white wine under acidic pH conditions. One step derivatization of nucleophiles has been realized directly in wines using 4-methyl-1,2-benzoquinone (4MeQ) as an electrophilic probe. Derivatization conditions considering probe concentration, pH, reaction time, MS ionisation conditions and adducts stability, were optimized using model solutions containing standard sulfur and amino compounds (GSH, Cys, HCys and Ser-Aps-Cys-Asp-Ser, Asp-Met, Met and Glu). Ultra-high-performance liquid chromatography coupled to a quadrupole-time of flight mass spectrometer (UHPLC-QqTOF-MS) analysis of up to 92 white wines from different cultivars (Chardonnay, Sauvignon and Semillon) followed by Multivariate analysis (PLS DA) and Wilcoxon test allowed to isolate up to 141 putative wine relevant nucleophiles. Only 20 of these compounds, essentially thiols, were detectable in samples before derivatization, indicating the importance of the quinone trapping on the revelation of wine unknown nucleophiles. Moreover, annotation using online database (Oligonet, Metlin and KEGG) as well as elementary formula determined by isotopic profile, provided evidence of the presence of amino acids (Val, Leu, Ile, Pro, Trp, Cys and Met) and peptides with important antioxidant properties. The complimentary set of MS/MS spectral data greatly accelerated identification of nucleophiles and enabled peptides sequencing. These results show that probing wines with 4-methyl-1,2-benzoquinone enhances thiols ionisation capacity and gives a better screening of specific S- N- containing functional compounds as part of the white wines antioxidant metabolome.

## 1. Introduction

Wines oxidative stability can be related to intrinsic and extrinsic factors preventing aroma deterioration. The majority of the studies dealing with antioxidant capacity of white wines have focused on the protective effect of targeted compounds, like sulfites, phenolics or glutathione (GSH) [1,2,3,4]. For a complex matrix like white wines, to establish the role of antioxidant naturally present is not easy, and to predict wine oxidative evolution using only targeted compounds can be insufficient. Our recent study based on the combination of FT-ICR-MS and multivariate statistical analyses has evidenced that GSH efficiency against wines sensory oxidative stability during bottle aging is dependent on wines global antioxidant metabolome consisting of N- and S-containing compounds like amino acids, aromatic compounds and peptides [5]. These compounds present a strong nucleophilic character and their reactivity with wines electrophiles such as oxidized polyphenols, suggests the formation of stable adducts presenting lower oxidative potential [6,7,8,9,10].

The analysis of thiol compounds is often difficult because of the reactivity of the sulfhydryl group, causing autoxidation, and low sensitivity related to their poor ionization level in electrospray mass spectrometry [11,12,13,14]. Therefore derivatization reagents are used to stabilize thiols and enhance their limit of detection [11,15]. The most used thiol derivatization reagents can be classified into five groups: disulfides, active halogens, aziridines, organic mercurial compounds and N-substituted maleimides [11,16,17,18,19]. However, these derivatization agents are intended for use in alkali pH conditions. The protonation state of the sulfhydryl group plays an important role in nucleophilic reaction efficiency, it is why the solution pH is important to derivatization yield [20]. Furthermore, nucleophilic derivatization under alkali conditions may cause side reactions in wine matrix, including the destruction of natural polyphenolic compounds, peptides, proteins and amino acids [21,22]. Indeed, the pH value in wine matrix controls first, the phenol-phenolate equilibrium, which sets the phenol concentration needed to undergo oxidation reactions in the presence of trace metals and oxygen [23,24]. At the higher pH range of wines (4.0) the concentration is about 10 times higher than at the lower pH range (3.0), and hence the oxidation rate of wines with high pH is much faster than at low pH [25]. Second, high pH values (≥ 7.0) suggest thiol containing compounds oxidation by reacting with the quinone formed by phenolic compounds oxidation. Jongberg et al. (2011) showed, in wine like acidic conditions, that amino and thiol containing compounds (like cys) react with 4-methyl-1,2-benzoquinone (4MeQ) forming a thiol-quinone adduct, where 4MeQ is reduced to diphenol form (4MeC) [6]. The reactivity of several wine relevant nucleophiles, like amino acids, volatile odorant thiols and antioxidant compounds against 4MeQ showed rapid reaction rates and yields up to 95% [6,8].

In most of the studies, known compounds were targeted for quantification, especially GSH, Cys and HCys [6,12,16,18,19,20]. However, the untargeted molecular analysis of sulfur compounds in wine requires either the coupling of chromatography for specific detection [26], or derivatization strategies prior to liquid chromatography–mass spectrometry, coupled with multivariate statistical analysis [17]. In that case, authors showed that the quality of the identification is directly related to the decrease of native nucleophiles and the increase of their derivatized forms. Recently, Ma et al. (2018) have used ^13^C labeled ortho-quinone to isolate putative nucleophilic compounds of wines by comparison with unlabeled ortho-quinone [27].

Here, our effort was aimed at investigating the performance of high-resolution mass spectrometry with multivariate statistical analysis as a tool for the direct analysis of wine relevant nucleophiles without matrix modification. In order to eliminate pH-related derivatization artifacts of existing protocols, 4MeQ was used as a strong electrophilic probe under wine acidic pH conditions. 4MeQ was also chosen as derivatization agent for its proven direct reactivity and affinity with wine nucleophilic compounds. Derivatization conditions were optimized based on UHPLC-QqTOF-MS profiling of known nucleophilic compounds. Extracted masses, related to the mass difference between identified 4MeQ derivatives and the known 4MeQ, associated to their retention times were submitted to metabolomic databases for a first identification confidence level. Second identification confidence level, related to structural characterization and peptides sequencing was given after MS/MS analysis. Thanks to our novel analytical approach, unknown nucleophilic compounds have been isolated and characterized in freshly made white wines from different grape varieties, giving a better understanding on wines antioxidant metabolome composition.

## 2. Experimental Section

### 2.1. Chemicals

MS grade acetonitrile was obtained from Biosolve (Dieuze, France), formic acid (MS grade) from Acros Organic (Morris Plains, NJ, USA), NaOH from ChemLab (Zedelgem, Belgium), ethanol from Fisher Chemical (Loughborough, UK), Tuning Mix from Agilent Technologies (Santa Clara, CA, USA), 4-methyl-catechol (4MeC), Amberlyst A-26(OH) ion-exchange resin, periodic acid, tartaric acid, cysteine (Cys), homocysteine (HCys), glutathione (GSH), glutamic acid (Glu) and methionine (Met) were purchased from Sigma Aldrich (St. Louis, MO, USA). Ser-Asp-Cys-Asp-Ser, Asp-Met were obtained from GeneCust (Dudelange, Luxembourg) and ultrapure water comes from a Milli-Q system (Merck, Darmstadt, Germany).

### 2.2. Wine Samples

92 young wines from three grapes varieties, Chardonnay (62 wines), Sauvignon (24 wines) and Semillon (6 wines), were analyzed over two vintages (2016 and 2017). Sixteen chardonnays came from the 2016 vintage and they were analyzed after ageing for one year on lees in barrel. The remaining 76 wine samples stem from the 2017 vintage and they were analyzed directly after alcoholic fermentation (Appendix A).

### 2.3. Derivatization using 4-methyl-1,2-benzoquinone

1 mL of wine or standard solution (GSH, HCys and Cys at 0.05 mmol/L, Glu and Met at 0.2 mmol/L, Ser-Aps-Cys-Asp-Ser at 9.5 µmol/L and Asp-Met at 19 µmol/L) in model wine (12% (*v/v*) ethanol in water, 6 g/L of tartaric acid and pH 3.2), were kept in dark under argon. 1 mM (50 µL) of 4-methyl-1,2-benzoquinone (4Me-Q) prepared in acetonitrile according the protocol proposed by Nikolantonaki and Waterhouse (2012) [8] were then added. After 30 min incubation, 1 mmol/L of sulfites was added to quench the reaction. Blank samples were prepared each time by adding acetonitrile and sulfites in equal concentrations as in the derivatized samples. Samples were analyzed within 24 h (sd < 1%).

### 2.4. Ultra-High-Performance Liquid Chromatography Coupled to a Quadrupole-Time of Flight Mass SpectrometerAnalysis

Analyses were realized using an ultra-high-pressure liquid chromatography (Dionex Ultimate 3000, Thermo Fischer Scientific, Waltham, MA USA) coupled to a MaXis plus MQ ESI-QqTOF mass spectrometer (Bruker, Bremen, Germany). The column used was an Acquity BEH C18 1.7 µm, 100 × 2.1 mm (Waters, Guyancourt, France) in reverse phase to analyzed non-polar compounds. The mobile phase was water +0.1% (*v/v*) of formic acid for eluent A and 95% (*v/v*) acetonitrile +0.1% (*v/v*) of formic acid for eluent B. The temperature of elution was 40 °C using the gradient: 0–1.10 min 5% (*v/v*) of eluent B and 95% (*v/v*) of eluent B at 6.40 min. The flow was set at 400 µL/min. The ionization took place in electrospray (2 bars pressure for nebulizer and 10 L/min for nitrogen dry gas flow) in positive ion mode. End plate offset (500 V) and capillary voltage (4500 V) permitted the ions transfer. To recalibrate spectrum, 4 times diluted calibrant ESI-L Low Concentration Tuning Mix (Agilent, Les Ulis, France) was injected at the beginning of each run. Before batch analysis, the mass spectrometer was calibrated using undiluted Tuning Mix in enhanced quadratic mode (errors <0.5 ppm). The mass range was between 100 and 1500 *m/z*. Quality controls were analyzed before and throughout each batch, to verify the stability of the UHPLC-QqTOF-MS system. All samples were analyzed randomly. Fragmentation has been realized on targeted compounds using 20 eV as collision energy. Limit of detection (LOD) and limit of quantification (LOQ) were calculated for tested nucleophilic compounds as LOD = $\frac{3*\left[C\right]}{S/N}$ and LOQ = $\frac{10*\left[C\right]}{S/N}$, where [C] is the concentration of compounds without derivatization and the signal on noise ratio (*S/N*).

### 2.5. Data Mining

Features (*m/z*, retention time) were filtered according to S/N higher than 30 and an absolute intensity above 1000. The spectral background noise was removed before features extraction. A homemade R script allowed alignment of extracted features with m/z and retention time tolerance lower than 10 ppm and 0.3 min, respectively. Partial least squares discriminant analysis (PLS-DA) were realized with Simca (Umetrics) to determine VIP scores for each molecular feature. Using a Matlab (R2015a) script, Wilcoxon correlations of each feature were calculated. Criterion was fixed to VIP > 1 and *p*-value_Wilcoxon_ < 0.01 allowing to keep statistical discriminant molecular features, which could be native or derivatized compounds. The elementary composition of the filtered compounds was determined using isotopic profile with DataAnalysis (v. 4.3, Bruker, Mannheim, Germany). Online tools and databases such as Oligonet [28], KEGG [29] and Metlin [30] were used for the putative annotation of derivatized compounds (mass detected—mass of 4MeQ). These information were compared with MS^2^ analysis to validate the elementary formula and the structure of the compound. Annotation confidence level has been calculated for each compound according to Hollender et al. (2014) and Trengove et al. (2014) [31,32].

## 3. Results and Discussion

### 3.1. Screening the Conditions of One-Step Derivatization of Model Wines for Optimizing the Detection of Relevant Nucleophiles in Acidic pH

Seven model wine relevant nucleophiles including free linear thiols (GSH, Cys, HCys and Ser-Aps-Cys-Asp-Ser), bound thiols (Asp-Met and Met) and amino compounds (Glu) presenting a molecular weight from 121 to 525 Da and different steric hindrances, were used as model compounds to optimize derivatization conditions.

An UHPLC-QqTOF-MS method was developed to separate the reaction products formed after the incubation of 4MeQ with, GSH, Cys, HCys, Ser-Aps-Cys-Asp-Ser, Asp-Met, Met and Glu in model wine at room temperature. The UHPLC–MS-QqTOF revealed the major addition products and several minor compounds (Table 1). After 4MeQ addition, the derivatized compounds were more hydrophobic permitting a better separation using C18 reverse phase liquid chromatography (Appendix A). A complete separation for all known, and for the future unknown analytes appears no longer necessary based on the high-resolution mass spectrometry detection mode in which analytes can be distinguished by their molecular weights. However, an excellent chromatographic separation would efficiently minimize matrix interference and remarkably enhance detection sensitivity.

The main compounds formed in model solutions were addition products including the single addition of one nucleophilic compound to one 4MeC skeleton according to the mechanism proposed by Nikolantonaki et al. (2012) (Scheme 1) [33]. In detail, three mono-adducts corresponding to [(4MeC+GSH-2H)+H]^+^, [(4MeC+Ser-Asp-Cys-Asp-Ser-2H)+H]^+^ and [(4MeC+Cys-2H)+H]^+^ and two mono-adducts corresponding to [(4MeC+Hcys-2H)+H]^+^ detected at the signal intensity ratio of 1:30:2 and 1:2, respectively, were detected based on their MS ionization peak patterns. Moreover, due to a low reactivity, only one monoadduct was detected after the reaction of 4MeQ with the tested amino compounds: Asp-Met, Met and Glu (Appendix A). For optimization conditions, we decided to focus on the predominant adducts detected in wines samples, and thus, parameters such as the concentration of the electrophilic probe (4MeQ), the pH value, the reaction time, the UHPLC separation conditions and the QqTOF-MS ionization conditions and the derivatization products stability were investigated in detail in model wine conditions.

#### 3.1.1. Electrophilic Probe (4MeQ) Concentration Effect

For ensuring completion and reproducibility of this one-step derivatization, an excess concentration of 4MeQ was necessary. The effect of 4MeQ concentration was investigated from 2- to 6-fold molar ratios excess to the total molar of nucleophilic compounds. The results suggested the highest and most stable peak intensity was achieved when the amount of 4MeQ was 4-fold molar excess to the total molar of nucleophiles. Under this condition, 4MeQ was in large excess, and the nucleophiles were completely probed.

#### 3.1.2. pH Effect

The reactivity of sulfhydryl and amino groups is highly dependent on the protonation state, which depends on the solution pH [20]. In that respect, the pH influence on nucleophilic reaction yields was studied in the range of 3.0–8.0 (Figure 1). Among tested nucleophiles, the pH had an important impact on the reaction. The peak areas of the probed thiol-containing derivatives (Nu = GSH, Cys, HCys or Ser-Aps-Cys-Asp-Ser) reached a maximum in the pH range 3.0–5.0. Interestingly, GSH-4MeC and Ser-Aps-Cys-Asp-Ser-4MeC presented a peak area maximum at pH 3.0, while pH 5.0 was optimum for Cys-4MeC and HCys-4MeC. Based on the p*Ka* values of Cys (p*Ka* 8.3) and HCys (p*Ka* 8.6) it is reasonable to assume that the reactivity of these thiols would be higher than that of GSH (p*Ka* 9.0), since a lower p*Ka* value would allow for higher thiolate dissociation (−S^−^), which possesses stronger nucleophilicity, and consequently higher affinity for the electrophilic probe 4MeQ. In acidic conditions, the overall picture is far more complex because p*Ka* is not the only contributing factor. In fact, the steric hindrance of a specific thiol molecule at a given pH has considerable influence in dictating the higher or lower affinity of electrophiles for thiols.

The effect of pH on nucleophilic addition to 4MeQ of amino containing compounds (Glu, Met and Asp-Met), including conjugated thiols, was linear, being minimum at pH 5.0 and maximum at the highest pH (8.0) (Figure 1). The yield of reaction of amines with carbonyls was very dependent on pH, with a yield increase with the pH increase from 5.0 to pH 7.0. Around pH 5.0 is the goldilocks point: acidic enough to increase the rate of reaction, but not too acidic. Below pH 5.0, the reaction yield started to decrease, which reflects the greater proportion of the conjugate acid of the amine, which is not nucleophilic. Below pH 3.0, no reaction was observed since all the free amine has been converted to the ammonium salt.

These results showed that pH had an important impact on the derivatization reaction yield of wine relevant nucleophiles with 4 MeQ. Indeed, low pH values (3.0–4.0) promote thiols probing, while higher pH (8.0) promotes amino compounds reactivity. With respect to our initial goal related to the estimation of the nucleophilic properties of wine matrices under real winemaking conditions, and the avoidance of side reactions under alkali conditions, it was decided to conduct derivatization reactions directly to wine samples without adjusting their pH (pH around 3).

#### 3.1.3. Reaction Time Effect

The reaction kinetic of tested nucleophiles with 4MeQ was evaluated at pH 3.0. Under our experimental conditions with the presence of 4 MeQ in 4-fold molar excess, reactions of all tested nucleophiles, except HCys and Met, were accomplished with yields higher than 90% in 30 min (Figure 2). In the case of HCys, the maximum reaction yield was only achieved after 200 min. However, under our experimental conditions and in accordance with the literature, Met-4 MeC was not detected [8]. Consequently, the derivatization reaction was carried out at room temperature for 30 min. After accomplishment, derivatization reactions were quenched by the addition of sulfites in excess.

#### 3.1.4. UHPLC-QqTOF-MS Ionization Conditions

The last parameter tested for the optimization of derivatization conditions was the ionization mode for their detection by mass spectrometry. Thiol and amino containing compounds can be ionized in positive or negative electrospray ionization (ESI) mode, but sulfur compounds are poorly ionized by ESI-MS [11,14]. However, chemical labeling can improve detection sensitivity with the introduction of an easily ionizable group into the targeted analytes. In this work, a model quinone (4MeQ) was used as a probe to target nucleophilic compounds. After nucleophilic addition, the quinone is reduced back to the phenol, which bears two hydroxyl groups that can be easily protonated in acidic conditions and thus improve the ionization efficiency of sulfur-containing nucleophilic compounds. Here, we used all tested nucleophiles to estimate the enhancement of the detection sensitivity upon 4MeQ derivatization. Table 1 shows that 4MeQ-derivatives clearly exhibited higher ionization potentials than native nucleophiles in both positive and negative modes. However, for the same concentration, GSH-4MeC, Cys-4MeC, HCys-4MeC and Ser-Aps-Cys-Asp-Ser-4MeC gave up to five times higher intensities in positive mode than in negative mode, whereas Asp-Met-4MeC and Glu-4MeC were detected only in positive mode. The LODs and LOQs of free and 4MeQ-derivatized nucleophiles, in positive ionization mode were gathered in Table 1. These results further confirmed the better ionization efficiency of 4MeQ-derivatives compared to the corresponding free forms of nucleophiles.

### 3.2. Multivariate Statistical Analysis of UHPLC-QqTOF-MS Data for the Isolation of Wine Relevant Nucleophiles

UHPLC-QqTOF-MS data were collected after derivatization of 92 wines from different varieties and vintages. The MS peaks from the scan chromatograms with and without derivatization samples were extracted, and the data were aligned (*m/z* values and retention times) using a home-made R script. The unsupervised statistical PCA of the complete dataset comprising 6723 features, from wines before and after derivatization did not deliver visual discrimination (Figure 3). This result demonstrated first, that derivatization conditions had no impact on the molecular fingerprint of wines, and secondly, that they were optimal, with the avoidance of any side oxidation reactions. Then, a PLS-DA model with Wilcoxon test (VIP > 1 and *p*-value < 0.01) was built to isolate statistically discriminant molecular features between features detected in wines with and without derivatization, and considered to be specifically associated with nucleophiles (Figure 4A). This data filtration step resulted in the identification of only 468 VIP molecular features, of which only 141 were specific to the derivatized wine samples (Figure 4B). As indicated on the basis of the difference *m/z* value of 122.0367 for the 4MeQ derivatization reaction among the 141 identified nucleophiles only 21 were detected in their free form directly in wines without derivatization. In that respect, this screening assay after one step derivatization facilitates the identification of unknown nucleophiles while promising a better characterization of wines antioxidant metabolome.

### 3.3. Annotation of Wine Relevant Nucleophilic Compounds

The 141 isolated *m/z* values were used for the questioning of online databases (KEGG and Metlin), along with the online tools Oligonet interface [28], permitting to annotate 84 molecular features. Molecular formulae were determined using first, the isotopic ratio pattern and second, the confidence annotation level estimated from two indexes proposed by Schymanski et al. (2014) and Summer et al. (2014) [31,32].

Table 2 gathers the putative identifications of the most abundant isolated compounds. Due to isomer formation after nucleophilic addition to three possible electrophilic carbon sites on the 4MeQ moiety, two (or three) molecular features with the same m/z but different retention times, could represent the same native compound. As example, molecular features detected with *m/z* 430.1278, 401.1372 and 471.1902 were all presented by two well resolved MS peaks at 2.06 and 2.35 min, 3.15 and 3.25 min and 2.84 and 3.12 min, respectively. However, the ratio of MS peak areas of isomers were specific to the compound and, due to low sensitivity MS^2^ experiments were conducted only for the major reaction product. A targeted MS^2^ mode was developed for these compounds with an optimized collision energy. Moreover, accurate mass data and isotopic distributions for the precursor and product ions were compared with theoretical and experimental spectral data of standard compounds when available.

MS^2^ analyses led to the identification of 26 out 141 isolated nucleophilic compounds. Identified compounds based on their fragmentation patterns were essentially sulfur containing amino acids and peptides. For example, the characteristic fragmentation of di-peptide, Cys-Gly, and tri-peptides, GSH and Val-Leu-Cys are shown in Figure 5. The major peaks at *m/z* 169.0914 and *m/z* 199.1800 in Figure 5A represent prompt fragments, resulting from the loss of 132.0 or 102.0 Da, respectively, from the protonated Cys-Gly-4MeC. Cys-Gly peptide sequence could be confirmed by the presence of the characteristic *m/z* 227.1712 fragment indicating the cleavage of the amine bond to produce b ion [34]. In addition, Figure 5B represents the MS^2^ spectrum of the precursor ion *m/z* 430.1278. The MS^2^ spectrum of the parent ion shows that major product ions have the following masses: 284.0586, 301.0851, 327.1009 and 355.0957. The main fragmentation pathways concern amino cleavages of Cys and Gly residues of the GSH moiety. While MS^2^ spectra comparison with this GSH-4 MeC standard compound confirmed structural identification. In our study, except GSH, Cys, Pro and HCys have been identified after matching of MS^2^ spectra with these of standards.

When standards were not available, MS^2^ spectra allowed to enhance the confidence into putative identification for certain compounds. Thus, the ions at *m/z* 357.1475, 456.2156 and 456.1431 which could be probably Leu-Cys (or Ile-Cys), Val-Leu-Cys (or Val-Ile-Cys) and Pro-Cys-Asp, respectively, all have a detected fragment at *m/z* 244.0637, which can be associated to Cys-QH_2_ (Table 2). Moreover, fragmentation pattern of the *m/z* 456.2170 precursor ion with major peaks at *m/z* 355.1324 and *m/z* 185.1645 clearly allowed peptide sequencing. In detail, *m/z* 355.1324 fragment with a mass loss of 101.0 Da, which corresponds to the Val (C_5_H_10_NO●) moiety after amine bond cleavage of the non-side chain (C1) carboxylic acid moiety of Val and the non-side chain (N2) amino moiety of Leu, while fragment at *m/z* 227.0314 corresponds to Cys-4MeC moiety (Figure 5C).

The fragmentation further allowed to enhance the probability of annotation of the Gly-Thr-Cys tripeptide (402.1329 *m/z*) due to 327.1306 m/z fragments detection which correspond to a loss a Gly fragment for the corresponding peptides. By the same mechanism, the presence of Val could be confirmed in Val-Ser-Cys (430.1638 *m/z*) and Val-Leu-Cys (458.1586 *m/z*), through the detection of fragments corresponding to a loss of Val, leading to respectively 313.1878 and 312.0898 *m/z*. Fragmentation also allowed to consolidate the annotation confidence of Cys-Glu (373.1063 *m/z*) thanks to the 227.1745 *m/z* fragment which is a loss of Glu amino acids.

Moreover, it appeared that certain amino acid combination seemed to be repeated among annotated compounds, such as Leu-Cys in Val-Leu-Cys, Leu-Cys-Asp, Gly-Leu-Cys and Gly-Gly-Leu-Cys and Cys-Asp in Leu-Cys-Asp, Val-Cys-Asp, Pro-Cys-Asp, which could be the result of bigger peptides degradation [28]. These repetitions clearly increased the pertinence of peptide identification. The derivatized compounds at *m/z* 472.1743 have two possible amino acids combination, Leu-Cys-Asp or Val-Cys-Glu. The presence of annotated peptides containing Leu-Cys and Cys-Asp increased the probability that this *m/z* 472.1743 contains Leu, Cys and Asp [35].

The annotation showed an important part of putative peptides in the isolated molecular features. It is known that in fermented food, antioxidant peptides can be form during fermentation [36]. These peptides can interact with free radicals thanks to their primary structures. The most reactive amino acids are sulfur containing amino acids (Cys and Met), the aromatic amino acids (Trp, Tyr and Phe), and the imidazole containing amino acids (His) [36,37]. In most cases, antioxidant peptides contain hydrophobic amino acids in the N-terminus chain (Val, Leu and Ile), and Pro, His, Tyr, Trp and Met in their sequence [36]. Annotation with Oligonet, allowed to determine peptides amino acids combination, with amino acids Val, Leu, Ile, Pro, Trp, Cys and Met. Theses peptides can contribute to antioxidant mechanisms in white wines by quinone trapping and/or free radical scavenging.

### 3.4. Grape cultivar Effect on Wines Antioxidant Metabolome

To question the grape cultivar effect on Chardonnay, Sauvignon blanc and Semillon nucleophilic composition related to their antioxidant metabolome a principal component analysis (PCA) was realized with wines from the 2017 vintage (Figure 6A, Appendix A). The PCA of the 46 Chardonnay, 24 Sauvignon blanc and six Semillon wine samples gave two discriminant groups among the cultivars (Group 1: Chardonnay and Group 2: Sauvignon blanc) on the component 2, while Semillon wines were not clearly discriminated. The 20 highest and 20 lowest variables on the second PCA component were then used to evaluate the molecular diversity of Chardonnay and Sauvignon blanc cultivars discriminant nucleophilic compounds. Among the annotated nucleophiles presented in Table 2, Cys (121.0197 Da), HCys (135.0359 Da) and Val-Ile-Cys (333.1722 Da) were more abundant in Chardonnay, while pantetheine was more present in Sauvignon blanc samples.

The semi-quantitative analysis of wine thiol-peptidome by means of a heat map was shown in Appendix A. No distinct differences in nucleophilic metabolite profiles among cultivars were clearly observed. In order to better understand the chemical diversity of the global antioxidant metabolome in a cultivar dependent manner, the mass-to-charge ratio (*m/z*) distribution of the most abundant nucleophiles was considered (Figure 6B). At the one hand, 17 thiol containing peptides with a mass distribution from 100 to 400 Da, which corresponds to tripeptides or dipeptides were discriminant for Chardonnay wines. On the other hand, Sauvignon blanc related peptides had a larger mass distribution from 100 to 700 Da, presenting 11 compounds at low (100–400 Da) and nine to high (400–700 Da) masses [38]. These results put in evidence the potential use of thiol-peptidome for discriminating complex wine matrices.

## 4. Conclusions

In this study, 4MeQ was used as derivatized agent to enhance the detection of thiols by UHPLC-QqTOF-MS and determine the nucleophilic pool of white wines. Optimal derivatization parameters were determined using standard nucleophiles in model wines, allowing to retain reaction time at 30 min, pH at 3 (no modification for wines samples) and confirming the adducts stability. This model study showed that at wine pH the most nucleophilic species are thiols, whereas amino compounds would only weakly react at this pH.

Multivariate analysis allowed to isolate untargeted nucleophilic compounds in up to 92 wines sample using PLS-DA and Wilcoxon test. 141 features were isolated, whereas only 21 native compounds were detected without derivatization, thus showing an important increase into nucleophilic compounds detection, especially for sulfur-containing compounds. Annotation by online databases and tools (Oligonet, Metlin and KEGG), and questioning elementary formula determined by isotopic profile and MS^2^ analyses allowed to identify an important pool of thiols compounds, in particularly peptides. Four compounds were identified by matching with a standard compound, including 3 thiols compounds GSH, Cys, HCys and an amino acid Pro. Due to their putative amino acid combinations (Val, Leu, Ile, Pro, Trp, Cys and Met), isolated peptide nucleophiles can play a role in the antioxidant mechanism of white wines by quinones trapping and free radical scavenging.

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
