# Peer review of "Molecular Characterization of White Wines Antioxidant Metabolome by Ultra High Performance Liquid Chromatography High-Resolution Mass Spectrometry"

_antioxidants, 2020, doi:10.3390/antiox9020115_

Round 1

Reviewer 1 Report

I recommend this paper should be polished up focusing on:

detailed explanation on comments, Objective conclusions based on results and is resubmitted

highlight the purpose of the study in Abstract In the Results and Discussion section are mostly results, with negligible discussion. The discussion should have this form: 1. the main results; 2. interpret why – mechanisms; 3. originality; 4. restrictions; 5. how it fits in with previous studies; 6. what can be further explored? An introduction provides sufficient background and include all relevant references. The research design is appropriate, but the methods aren't adequately described, and the results aren't clearly presented. The conclusions are generally supported by the results, but the whole article has an insufficient emphasis on the antioxidant mechanisms in white wines for publication in Journal

Author Response

This letter consists to a document replying in green each point raised to the revised manuscript

R1

I recommend this paper should be polished up focusing on:

detailed explanation on comments, Objective conclusions based on results and is resubmitted

In the Results and Discussion section are mostly results, with negligible discussion.

The discussion should have this form: 1. the main results; 2. interpret why – mechanisms; 3. originality; 4. restrictions; 5. how it fits in with previous studies; 6. what can be further explored?

An introduction provides sufficient background and include all relevant references. The research design is appropriate, but the methods aren't adequately described, and the results aren't clearly presented.

The conclusions are generally supported by the results, but the whole article has an insufficient emphasis on the antioxidant mechanisms in white wines for publication in Journal

We thank reviewer 1 for his/her comments, however, we do not understand the arguments about the form of the introduction, the results and discussion section and the conclusion!? The aim of this paper is to propose an original characterization of a molecular fraction that contributes to white wines antioxidant properties, and not to characterize the various antioxidant mechanisms involved. Yet, and in agreement with comments from the other reviewers, we have added some sentences in the introduction, the discussion and conclusion.

Reviewer 2 Report

Dear Sir or Madam,

the manuscript „ Molecular Characterization of White Wines Antioxidant Metabolome by Ultra High Performance Liquid Chromatography High-Resolution Mass Spectrometry “ describes a new analytical approach for vine analysis by HR-MS after derivatization of nucleophylic metabolites with 4-methyl-1,2-benzoquinone. This manuscript can be published in Antioxidants after major revision, i.e. addressing the following aspects:

Major remarks

English requires improvement. It is seen already at the 4th line of the abstract – in English it would be “lipid and polyphenol oxidation”. The authors should arrange proof reading. Line 20 and as applicable: please, explain abbreviations by the first use Line 48, 69: very heavy mistake – the field of knowledge “mass spectroscopy” does not exist. For me it indicates poor understanding of the methodology by the authors. The authors need to proof read the manuscript with a person familiar with mass spectrometry. Line 104 and everywhere as appropriate: please express molar concentration according SI, i.e. as mol/L, but not M. Correct everywhere please. Lines 163-165: no conclusions about the relative contents of different analytes can be done on the basis of their MS intensity ratios!!! Ionization efficiency is substance-dependent! In general, I would somehow point out, that the authors target thiol-peptidome of wine, i.e. small cys-containing peptides. It looks to be an interesting output The manuscript would profit much, if the relative quantitative analysis of identified metabolites over the whole set of wines would be provided

Minor remarks

This “UPLC-QToF-MS” is 2x wrong: the chromatographic technique is ultra-high performance liquid chromatography, and it is abbreviated as UHPLC. You mass spectrometer is called “quadrupole-time of flight mass spectrometer” – it contains TWO quadrupoles, one filter Q and one RF-only multipole collision cell q. Therefore this machine is abbreviated QqTOF. Line 51: derivatization agents Line 111: see minor 1 and major 3. Line 117: write “eluent B” please. Line 123: actually it is called “positive ion mode” Lines 125-126: what collision gas settings were used? What gas and what pressure? Line 128: m/z is given in italic Line 130: statistical programming environment „Elementary formula“ – replace with elementary composition Table 1: somewhere needs to be explained, how LOD and LOQ were determined. Please , provide also linear dynamic range Line 168: what is “most sensitive” adduct? Sensitive to what? Less stable? What then the logic of the following text? Please, clear this Scheme 1: not really convenient with XIC – try to show it somehow more in the way, attracting focus. The labels +1 at the spectra are senseless for a specialist – they already see that you show single charged signals. Remove please Figure 1: what “arbitrary units” are meant here? MS produces data in counts. Lines 207-209: something is wrong here – it does not correspond to Figure 1 and its legend. Figure 4: according to the figure, in the legend must be “with and without derivatization”, but not “before and after” Line 277: Metlin?

Reviewer 3 Report

The manuscript entitled “Molecular characterization of white wines antioxidant metabolome by ultra high performance liquid chromatography high-resolution mass spectrometry” evaluated the performance of ultra high-resolution mass spectrometry with multivariate statistical analysis as a tool for the direct analysis of wine relevant nucleophiles without matrix modification. Moreover, in the current study the derivatization conditions such as concentration, pH, reaction time, MS ionisation conditions and adducts stability, were optimized using model solutions containing standard sulfur and amino compounds. In my opinion, this manuscript should be accepted to be publish in Antioxidant after minor revision.

All abbreviations should be described when used for the first time, and then adopted through the manuscript. Other abbreviation should be used to represent 4-methyl-1,2-benzoquinone, since Q is the abbreviation for Quadrupole.

The authors should standardized the units sometimes appears g/L others g.L-1.

Line 108: “…30 mn..” should be “30 min…”

Figures 1 and 2. Please adjust the axis values to data, in order to clarify the interpretation.

Line 245: “The limits of detection (LODs) and quantification (LOQs)…” should be “The LODs and LOQs…”

Line 352: “…LC-QTOF-MS..” should be “…. UPLC-QToF-MS”

Author Response

This letter consists to a document replying in green each point raised to the revised manuscript

R3

The manuscript entitled “Molecular characterization of white wines antioxidant metabolome by ultra high performance liquid chromatography high-resolution mass spectrometry” evaluated the performance of ultra high-resolution mass spectrometry with multivariate statistical analysis as a tool for the direct analysis of wine relevant nucleophiles without matrix modification. Moreover, in the current study the derivatization conditions such as concentration, pH, reaction time, MS ionisation conditions and adducts stability, were optimized using model solutions containing standard sulfur and amino compounds. In my opinion, this manuscript should be accepted to be publish in Antioxidant after minor revision.

All abbreviations should be described when used for the first time, and then adopted through the manuscript. Other abbreviation should be used to represent 4-methyl-1,2-benzoquinone, since Q is the abbreviation for Quadrupole.

The authors should standardized the units sometimes appears g/L others g.L-1.

Line 108: “…30 mn..” should be “30 min…”

Figures 1 and 2. Please adjust the axis values to data, in order to clarify the interpretation.

Line 245: “The limits of detection (LODs) and quantification (LOQs)…” should be “The LODs and LOQs…”

Line 352: “…LC-QTOF-MS..” should be “…. UPLC-QToF-MS”

We thank reviewer 3 for his/her constructive comments, which have been carefully considered.

In particular:

- Abbreviations have been described when used for the first time, and 4-methyl-1,2-benzoquinone has been abbreviated as 4MeQ, and its reduced form as 4MeC.

- Units standardized accordingly.

- Axes of Figure 1 and 2 adjusted

Round 2

Reviewer 2 Report

Dear Sir or Madam,

the authors have inproved the text, the manuscript became much more sounding.

A couple of questions are still opened:

The statement in the lines 171 - 174 is WRONG! You can not tell abouth the ratio of analytes based on the ratio of their intensities. Replace "in a ratio" by smth like "detected at the signal intensity ratio of ..." The authors should check the history of the QqTOF technology. Who invented it, how named and patented. The names of mass analyzers and trade names is not the same. And if many people write QTOF, QToF and so on, it does not mean that it is correct. It just means, that they don't know the thing, and the reviewers may be also...Please change name of the platform to QqTOF.

This needs to be corrected before the manuscript can be accepted.

For the outher issues with - Figure 1 and Scheme 1 I woul accept the answers of the authors.
